METHODS AND RESOURCES

# tACS entrains neural activity while somatosensory input is blocked

**Pedro G. Vieira**[ID][☯]**, Matthew R. Krause**[ID][☯]**, Christopher C. Pack***

Department of Neurology and Neurosurgery, McGill University, Montreal, Quebec, Canada

☯ These authors contributed equally to this work.
* christopher.pack@mcgill.ca

**Data Availability Statement:** All relevant data needed to recreate the figures and analysis are provided as Supporting Information. Please contact the corresponding author for access to the raw wideband signals.

## Abstract

Transcranial alternating current stimulation (tACS) modulates brain activity by passing electrical current through electrodes that are attached to the scalp. Because it is safe and noninvasive, tACS holds great promise as a tool for basic research and clinical treatment. However, little is known about how tACS ultimately influences neural activity. One hypothesis is that tACS affects neural responses directly, by producing electrical fields that interact with the brain's endogenous electrical activity. By controlling the shape and location of these electric fields, one could target brain regions associated with particular behaviors or symptoms. However, an alternative hypothesis is that tACS affects neural activity indirectly, via peripheral sensory afferents. In particular, it has often been hypothesized that tACS acts on sensory fibers in the skin, which in turn provide rhythmic input to central neurons. In this case, there would be little possibility of targeted brain stimulation, as the regions modulated by tACS would depend entirely on the somatosensory pathways originating in the skin around the stimulating electrodes. Here, we directly test these competing hypotheses by recording single-unit activity in the hippocampus and visual cortex of alert monkeys receiving tACS. We find that tACS entrains neuronal activity in both regions, so that cells fire synchronously with the stimulation. Blocking somatosensory input with a topical anesthetic does not significantly alter these neural entrainment effects. These data are therefore consistent with the direct stimulation hypothesis and suggest that peripheral somatosensory stimulation is not required for tACS to entrain neurons.

## Introduction

Recent results suggest that transcranial alternating current stimulation (tACS) can noninvasively alter brain activity [1–4], but the physiological mechanisms behind these exciting findings remain poorly understood. Traditionally, tACS is thought to produce oscillating electric fields within the brain that hyperpolarize and depolarize neurons, so that they fire synchronously with the stimulation. Small-animal experiments demonstrate that the fields generated by applying current to the bare skull can entrain neurons [1, 4, 5], consistent with the idea that intracranial electric fields have a direct effect on brain activity. In humans, however, the tACS electrodes are placed on the participant's intact scalp, not within the skull. Since the skin is

**Funding:** This work was supported by Defense Advanced Research Projects Agency Contract N66001-16-C-4058 (Memory Enhancement by Modulation of Encoding Strength; to CCP), Canadian Institutes of Health Research Grant MOP-115178 (to CCP), and a Jeanne Timmins Costello Postdoctoral Fellowship (to PGV). The funders had no role in study design, data collection and analysis, decision to publish, or preparation of the manuscript.

**Competing interests:** The authors have declared that no competing interests exist.

**Abbreviations:** EEG, electroencephalogram; LFP, local field potential; PLV, phase-locking value; tACS, transcranial alternating current stimulation; TOST, two one-sided tests.

highly conductive, but the skull beneath is not, much of that current is shunted away from the brain and stimulates neurons in the skin instead [6]. Rhythmic activation of somatosensory fibers could thus indirectly entrain central neurons by providing them with temporally structured sensory input. Since shunting also weakens electric fields in the brain, this indirect mechanism has been frequently proposed to be the dominant mode of action in humans [5, 7–10]. If this were true, it would have dramatic implications for how tACS is used and studied: brain areas would need to be targeted on the basis of somatosensory connectivity, rather than physical location, and brain regions that received little or no somatosensory input would be unreachable.

These competing hypotheses can be distinguished through the use of topical anesthesia. Pretreatment of the skin under and around the tACS electrodes with topical anesthetic blocks cutaneous afferents [11] and prevents them from generating somatosensory percepts [12]. If tACS acts indirectly via somatosensory inputs, topical anesthesia should reduce or abolish its effects by blocking transmission from the periphery. Conversely, if the electric fields directly affect neurons, applying topical anesthesia should produce little or no changes in the effects of tACS, as the electric fields produced within the brain remain the same. Previous attempts to test the indirect hypothesis have used proxy measurements for neural activity, with mixed results: topical anesthesia appears to prevent tACS from affecting nociception [10] and tremor [5], but effects on motor-evoked potentials [13] and language processing [14, 15] persist when somatosensory inputs are blocked. Interpreting these results is challenging, because the neural mechanisms behind these readouts are not well understood and each may involve multiple brain regions, only some of which may have been affected by the tACS used in each study.

An unambiguous test of the role of somatosensory input is to directly measure neural entrainment during tACS, with and without topical anesthetic. Here, we perform that decisive experiment in nonhuman primates, a highly realistic model for human neurostimulation. Using single-unit recordings of neurons in the hippocampus and visual cortex, we find that blocking somatosensory input has little effect on neural entrainment by tACS. Instead, our data support claims of a direct effect on neurons in the stimulated regions.

## Results

We collected data from two adult male rhesus monkeys (*Macaca mulatta*), using techniques and experiments that, with the exception of the topical anesthesia, are virtually identical to those in our previous work [3, 16]. Monkey N (7 year old male, 10 kg) participated in the experiments described in our previous study [3], whereas Monkey S (9 year old male, 20 kg) was obtained specifically for these experiments. These procedures were approved by the Montreal Neurological Institute's Animal Care Committee and conformed to the guidelines of the Canadian Council on Animal Care.

First, we determined if 5% EMLA cream, a widely used topical anesthetic, blocked the somatosensory stimulation produced by tACS. We used the animal's behavioral performance to validate the effectiveness of this anesthesia. In our previous studies [3, 16], animals were initially distracted by the onset of tACS but eventually learned to continue working despite the evoked percepts. If EMLA effectively blocks somatosensation, applying it around tACS electrodes should reduce these distractions and increase the time spent on task. Since a naive subject would be most sensitive to these effects, Monkey S, a well-trained monkey that had never received tACS, was tested using the paradigm in Fig 1A. Two pairs of tACS electrodes were placed on its head, one on anesthetized skin over one hemisphere and the other at identical locations on the untreated contralateral side. After a 20-minute delay, introduced to account

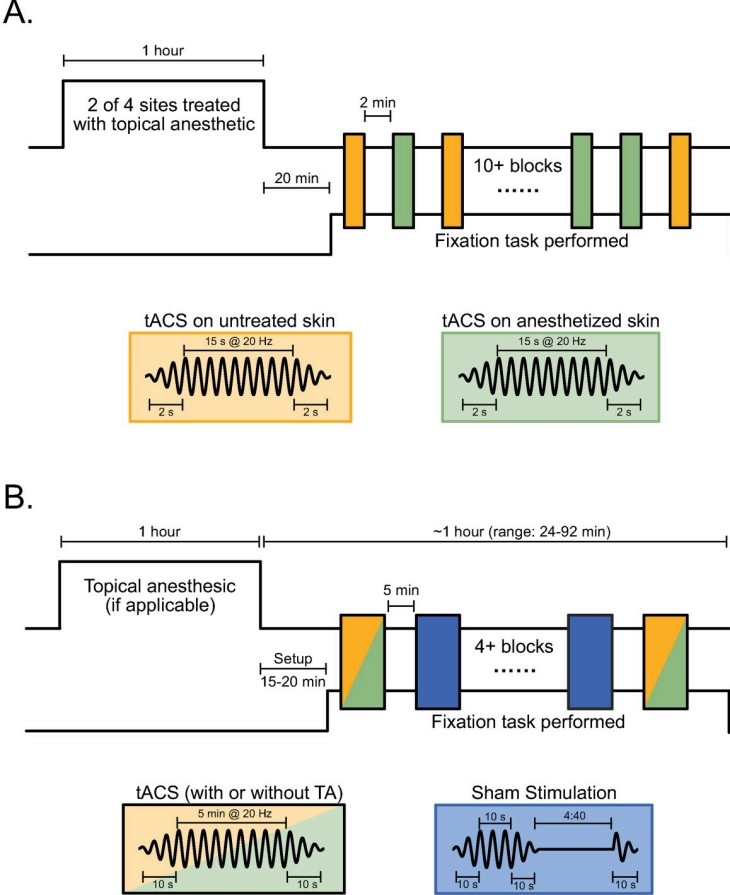

**Fig 1. Experimental design. (A)** To measure the effectiveness of topical anesthesia, we applied short bursts of tACS through two pairs of electrodes, one of which was placed over skin pretreated with 5% EMLA cream (green); the skin under the other pair was left untreated (yellow). Since subjects typically adapt to tACS sensations, the short bursts and brief ramps maximize any behavioral effects of stimulation. **(B)** To record neural responses to tACS, stimulation was applied through a single set of electrodes placed at scalp locations that optimally stimulated left hippocampus or left V4. In some sessions, the skin beneath both electrodes was pretreated with topical anesthetic (green); in others, the skin was left untreated as a control (yellow). In every session, a mixture of active tACS (yellow or green) and sham stimulation (blue), was applied for 5 minutes, with the ramps increased to 10 seconds to reduce sensations. This design ensures that identical electric fields were generated during the topical anesthesia and control conditions. In both experiments, blocks consisting of both conditions (pseudorandomly ordered) were repeated while the monkey continued to work, up to a maximum of 90 minutes. The tACS frequency was 20 Hz with amplitudes of ±0.5, ±1, or ±2 mA (i.e., 1, 2, or 4 mA peak to peak). Inset figures show the tACS waveforms in each condition. tACS, transcranial alternating current stimulation.

for setup time needed in subsequent neurophysiology experiments, the monkey performed a visual fixation task while bursts of tACS were applied through each pair of electrodes.

As Fig 2 shows, time on task was significantly increased ($p < 0.05$; Wilcoxon sign-rank tests) at all stimulation amplitudes when tACS was applied to the anesthetized skin, as compared to control sites. Furthermore, median performance was near ceiling (100% time on task) for ±0.5 and ±1 mA stimulation on anesthetized skin, but below 80% when stimulating at higher intensities or over control sites. These data suggest that EMLA increases somatosensory thresholds to approximately ±1 mA and reduces—but does not totally eliminate—sensation at ±2.0 mA, just as it does in humans [5]. We therefore collected neural data using currents that were both at perceptual threshold (±1 mA) and above it (±2 mA), reasoning that if

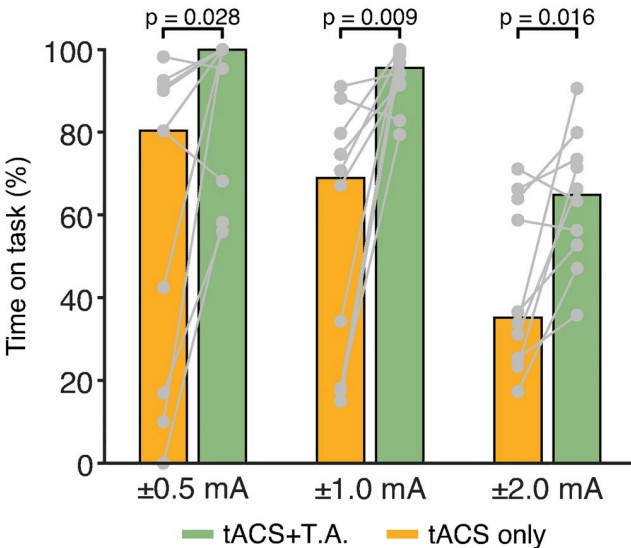

**Fig 2. EMLA effectively blocks or reduces tACS-related somatosensation.** In humans, tACS produces nociceptive sensations that disrupt behavior. The paradigm in Fig 1A was used to quantify these effects in our monkey subjects. We recorded the proportion of time that the animal's gaze remained within 2 degrees of the fixation target. T.A. (green bars) significantly reduced behavioral disruptions at all current levels when compared to stimulation applied over intact skin (yellow bars). Individual data points are shown in gray, and points from the same block are connected via solid lines; colored bars indicate the median. See S1 Data for individual values of each data point. T.A., topical anesthesia; tACS, transcranial alternating current stimulation.

somatosensory input were producing neuronal entrainment, reducing it either partially or completely with topical anesthetic would lead to a corresponding decrease or elimination of entrainment.

To test this hypothesis, we recorded neural activity with and without somatosensory blockade. During some recording sessions, the skin under and around each tACS electrode was pretreated with 5% EMLA; in others, the skin was left untreated as a control. This between-sessions design ensures that identical electric fields are produced during anesthesia and control conditions. In each session, interleaved 5-minute epochs of 20-Hz tACS and sham stimulation (Fig 1B; see Methods) were applied to the monkeys' scalps. For each neuron, we calculated two phase-locking values (PLVs): one summarizing its entrainment to the tACS waveform and another quantifying its entrainment to the matching frequency component (i.e., 20 Hz) of the local field potential (LFP) during sham [3]. These data allow us to determine the proportion of neurons that become entrained by tACS during each anesthesia condition. Comparing the strength of entrainment (PLVs) across tACS conditions provides an additional measure of the effects of topical anesthesia.

We first obtained data from the hippocampal recording sites described by Krause and colleagues (2019) [3]. As in that study, the tACS electrode montage was optimized to stimulate the left hippocampus, so that a ±2 mA alternating current produced an electric field of approximately 0.3 V/m at the recording site. Under control conditions (Fig 3A, yellow circles, $N = 8$ sessions), tACS entrained 50% of the hippocampal neurons ($N = 28/56$; $p < 0.05$ per-cell permutation tests). The median PLV during tACS was 0.054 (95% CI of the median 0.032–0.068), significantly larger than that observed during sham epochs (median: 0, CI 0–0.006; $p < 0.001$, $Z = -5.93$, Wilcoxon sign-rank test). Similar results were obtained from the 14 sessions where topical anesthetic was applied (Fig 3B, green circles): 45% of the hippocampal neurons ($N = 31/69$) were entrained by tACS, and PLVs significantly increased compared to sham

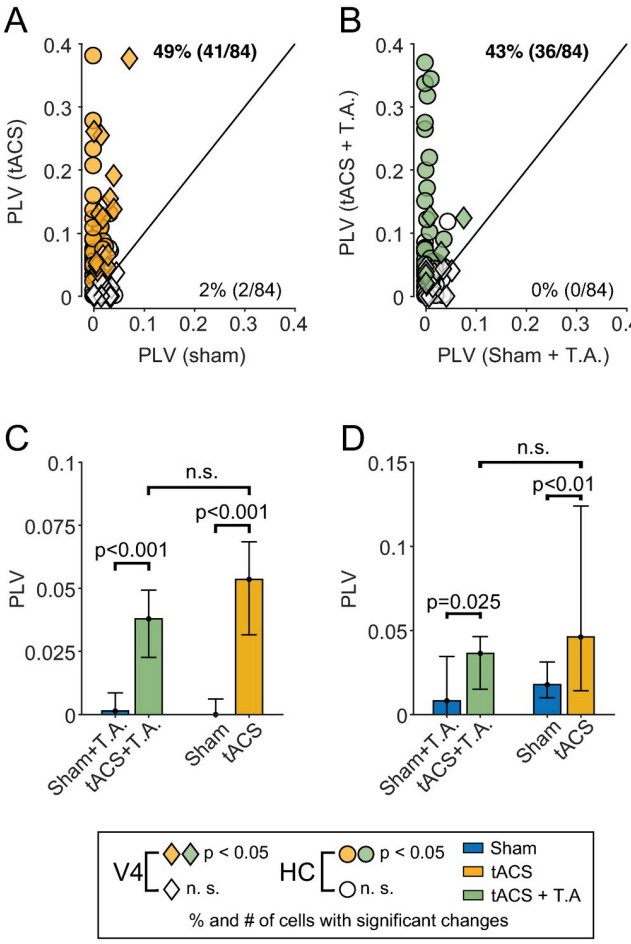

**Fig 3. Topical anesthesia does not reduce neuronal entrainment. (A and B)** PLVs for individual neurons recorded during control (A, yellow) and topical anesthesia sessions (B, green). Each point compares one neuron's entrainment during sham stimulation (x-axis) and tACS (y-axis). Circles and diamonds indicate hippocampal and V4 neurons, respectively. Filled shapes indicate neurons that exhibited an individually significant change ($p < 0.05$; randomization test) in entrainment between sham and tACS. PLVs tend to be above the unity line (black), indicating increased entrainment during tACS. The proportion of cells exhibiting significantly increased or decreased entrainment is shown above/below the unity line. Values in bold indicate proportions larger than would be expected by chance (i.e., outside the 95% binomial confidence interval for $p = 0.05$). **(C and D)** Summary plot of the data shown above. Bars indicate median and 95% confidence intervals of median for the data from each condition in the HC (C) and V4 (D). See S2 Data for values of each data point. HC, hippocampus; n.s., not significant ($p > 0.05$); PLV, phase-locking value; T.A., topical anesthesia; tACS, transcranial alternating current stimulation.

($p < 0.001$; Wilcoxon sign-rank test; Z = −5.39). Crucially, topical anesthesia did not significantly alter the strength of entrainment during tACS (Fig 3C; $p = 0.35$; Z = −0.92; Wilcoxon rank-sum test) or the proportion of neurons entrained ($p = 0.72$, odds ratio = 1.19; Fisher's exact test). An epoch-by-epoch regression analysis (see Methods section Topical Anesthesia) indicates that these results cannot be explained by a decrease in the effectiveness of the anesthesia over time.

Since these negative results may reflect true equivalence between the control and anesthetic conditions or a lack of statistical power, we performed equivalence testing [17]. We found that the difference between the proportion of entrained neurons is significantly equivalent to zero (within ±20%; $p = 0.047$, Z = −1.67; two one-sided tests [TOST]), suggesting that the results are due to equivalence between the conditions rather than a lack of statistical power. Similar

results were obtained when we lowered the stimulation amplitude to ±1 mA, below the animal's perceptual threshold (Fig 2). As S1 Fig shows, neurons tested during EMLA anesthesia still show significantly increased entrainment during ±1 mA tACS ($p < 0.05$; $N = 12$, randomization tests), and the strength of entrainment is again not significantly different between EMLA and control sessions ($p > 0.05$; Wilcoxon rank-sum test).

The hippocampus receives input from many brain regions and may therefore be especially sensitive to any residual somatosensory input not blocked by the topical anesthesia. We therefore recorded from 43 neurons in area V4, a midlevel visual area on the cortical surface that is less coupled to the somatosensory system [18–20]. For these experiments, the electrode montage was switched to one that optimally stimulated V4, and the current was maintained at ±1 mA to limit the resulting field strength to 1 V/m, which approximates that measured in human cortical areas [21, 22]; this stimulation amplitude also maintains skin stimulation below the animal's behavioral threshold. As before, tACS increased both the number of neurons entrained and the strength of their entrainment. Under control conditions, tACS entrained 46% of V4 cells (Fig 3A, diamonds, $N = 13/28$; $p < 0.05$ per-cell permutation tests) and led to a significant increase in PLV (median sham: 0.017, CI 0.009–0.031; median tACS: 0.046, CI 0.014–0.12; $p < 0.01$; $Z = −3.10$, Wilcoxon sign-rank test). Data collected during topical anesthesia sessions (Fig 3B, diamonds) were similar, with 33% ($N = 5/15$) of the neurons showing increased entrainment to the tACS and a statistically significant increase in PLVs during tACS as compared to sham stimulation (median sham: 0.008, CI 0–0.035; median tACS: 0.036, CI 0.015–0.046; $p = 0.025$; $Z = -2.23$, Wilcoxon sign-rank test). Entrainment during EMLA sessions was, again, not significantly different, in terms of median PLV value ($p = 0.36$; $Z = -0.92$; Wilcoxon rank-sum) or proportion of neurons entrained ($p = 0.52$, odds ratio = 1.71, Fisher's exact test) between these two conditions (Fig 3D). These results are generally inconsistent with the indirect stimulation hypothesis of tACS.

Examining the stimulation phase at which neurons become entrained provides an alternative method for distinguishing between direct and indirect effects [23]. During tACS, the preferred firing phases of individual neurons are significantly concentrated around 90˚ ($V$-test: $p = 0.0121$ for $\Theta_0 = 90$˚). As shown in S2 Fig, this effect is not present during sham ($p = 0.75$). Since the electric field's depolarizing influence is strongest in both the skin and brain at 90˚, where the tACS waveform peaks, this is consistent with a direct effect on neurons within the stimulated area. In contrast, indirect peripheral influences would likely arrive at our recording sites later, because of conduction and synaptic transmission delays. Only a handful of intervening synapses, each with a few milliseconds' delay [24, 25], would be sufficient to introduce a lag of 45˚ (6.25 ms) in the preferred phase. A very precise combination of lower firing thresholds in skin and specific conduction delays could reproduce this finding, but this remarkable coincidence could only occur at a specific tACS frequency, which is difficult to square with our previous results [3] demonstrating a shift toward 90˚ at frequencies ranging from 5 to 40 Hz.

## Discussion

These results demonstrate that neuronal entrainment by tACS can survive a topical blockade of somatosensation; indeed, for our recording sites, there was little discernible effect of blocking peripheral somatosensory inputs. These data argue strongly against an indirect account of tACS based on entrainment of somatosensory afferents. Retinal afferents have also been proposed as a possible route for indirect sensory entrainment. However, we previously showed that a cortical visual area physically distant from the stimulation target was not entrained by tACS, suggesting that retinal input also does not produce the observed effects [3]. Likewise, stimulation targeting the contralateral hemisphere, but producing similar retinal and

somatosensory input, failed to entrain neurons at our recording site [3]. Taken together, these results are more consistent with a direct effect on the spike timing of central neurons.

Previous work has attempted to address the role of somatosensory input by varying the locus of stimulation. We previously reported that entrainment of hippocampal and basal ganglia neurons was abolished by shifting the tACS electrodes to the contralateral hemisphere [3]. Likewise, Johnson and colleagues [2] found that shoulder stimulation did not entrain neurons in the pre- and postcentral gyrus. However, Asamoah and colleagues [5] reported that transcutaneous stimulation of the limbs entrained neurons and electroencephalogram (EEG) in motor cortex. Variations in somatosensory innervation or connectivity between test and control stimulation montages have been suggested as a possible explanation for these discrepancies [26], but our experiments allow us to exclude that mechanism by stimulating the same skin locations, with and without somatosensory input.

These experiments used field strengths that are representative of human tACS. Fields of up to 0.8–1.0 V/m have been predicted and measured in human cortex [21, 22], and even stronger fields (of up to 2 V/m) may be achievable in humans by using a multielectrode stimulation montage [27]. With the two-electrode montages used here, fields are strongest near the cortical surface, so other areas may receive stronger stimulation than our hippocampal target. As briefly discussed in [28], without recording from a multitude of other cortical areas, it is not possible to rule out indirect entrainment of hippocampal neurons via the cortex, but our results suggest that indirect somatosensory effects from the periphery are unlikely. This is even less likely in V4, a midlevel visual area on the cortical surface that predominately receives visual inputs [18–20]. The resulting changes in spike timing are similar to those reported in conjunction with therapeutically relevant changes in human and animal behavior [3], suggesting that non-sensory effects of tACS may provide an effective method for adjusting humans' mental states.

In these experiments, we applied topical anesthesia to a 5-cm (diameter) circle of skin under and around each electrode. Since skin is highly conductive, current could spread to peripheral nerve fibers outside that area, albeit not to an extent that affects behavior (Fig 2). Stimulation of larger fibers, including cranial nerves, may be a particular concern, as they are deeper in the skin and therefore less affected by topical anesthesia. In fact, this mechanism may account for some of the previously reported somatosensory effects. In [5], the control stimulation that was applied to the forelimb skin may also drive the medial or radial nerves [29]; stimulation of these fibers has been reported to reduce tremor in humans [30]. That said, there are reasons to believe that cranial nerve stimulation may not be a major factor in our data. To the extent that the high conductivity of skin allows distant cranial nerves to be stimulated, it should also make any resulting effects relatively insensitive to the position of the electrodes. However, the effects of tACS appear to depend on the precise locations of the stimulating electrode. In our previous study [3], shifting electrodes onto the contralateral hemisphere (approximately 3–4 cm) was sufficient to completely abolish entrainment, even though the electric fields in the skin were similar in both cases. A number of experiments have also demonstrated that specific electrode montages are required to produce the hypothesized behavioral effects [31, 32]. Peripheral and central stimulation also appear to have synergistic effects, suggesting the existence of separate mechanisms [33]. Finally, as with somatosensory input, it remains unclear how stimulation of the cranial nerves would alter the timing of spiking activity in synaptically distant area V4 or the hippocampus. However, given the diverse phenomena associated with vagal nerve activity [34], this possibility cannot be discounted altogether.

Peripheral nerve stimulation nevertheless has undeniable perceptual consequences that may confound behavioral experiments and could produce neural effects in areas that receive

especially strong somatosensory input. The combined effects of sensory input, nerve fiber stimulation, and direct neuronal polarization will necessarily vary across stimulation protocols, brain regions, and behavioral tasks, frustrating any simplistic attempt at interpretation. Nevertheless, our data support the hypothesis that tACS entrains central neurons directly rather than via sensory input.

## Methods

This paper describes data collected from two adult male rhesus monkeys (*M. mulatta*), using techniques and experiments that, with the exception of the topical anesthesia, are virtually identical to those in our previous work [3, 16]. Monkey N (7 year old male, 10 kg) participated in the experiments described by Krause and colleagues [3] whereas Monkey S (9 year old male, 20 kg) was obtained specifically for these experiments.

### Ethics statement

All procedures were approved by the Montreal Neurological Institute's Animal Care Committee (#5031), conformed to the guidelines of the Canadian Council on Animal Care, and were supervised by qualified veterinary staff. When not in the laboratory, animals were socially housed, received regular environmental enrichment, and had access to large play arenas.

### Transcranial alternating current stimulation

Using the method described in our previous work [3, 16, 35], we built an individualized finite-element model from MRIs of the animals' head and neck, which were solved to find a two-electrode montage that maximized field strength at the recording sites. For the hippocampal recording sites, this montage corresponds to Fp1/O1 in 10–10 coordinates and was predicted to produce a field of 0.26 V/m when ±2 mA of current was applied. We verified this prediction in Krause and colleagues [3] and reported that the mean field strength was 0.19 ±0.02 V/m (mean ± standard error) with peak strengths of up to 0.35 V/m. For the V4 site, a montage consisting of Fp1/P7 was found to produce field of approximately 1 V/m with ±1 mA of stimulating current, which approximates field strengths achievable in humans [22].

We applied stimulation using an unmodified StarStim8 system (Neuroelectrics, Barcelona, Spain), using 1 cm (radius) high-definition Ag/AgCl electrodes (PISTIM; Neuroelectrics; Barcelona, Spain) coated with conductive gel (SignaGel) and attached to the intact scalp with a thin layer of silicon elastomer (Kwik-Sil, World Precision Instruments). Electrode impedance was continuously monitored during the experiment and was typically between 1 and 2 KΩ, and always below 10 KΩ.

Stimulation consisted of a 20-Hz sinusoidal waveform. In our previous experiments [3], we found that tACS at this frequency entrained about half of the hippocampal neurons; similar results were also obtained with 5-, 10-, and 40-Hz stimulation. Since human participants also report similar sensations when receiving tACS at frequencies between 2 and 64 Hz [36], we expect the neural and somatosensory aspects of our study will generalize to most tACS protocols.

For the neurophysiology experiments, current was linearly ramped up from 0 over 10 seconds, held at full amplitude (±1 or ±2 mA; i.e., 2 or 4 mA peak to peak) for 5 minutes, and then ramped back down to 0, again over 10 seconds. The sham stimulation contains the same ramp-up period, but current remained at full amplitude (±1 or ±2 mA) for 10 seconds, before being ramped back down. Since steeper ramps produced stronger percepts, the ramp length was decreased to 2 seconds for the behavioral experiments.

## Topical anesthesia

Since previous work has used 5% EMLA cream [5, 12, 13, 31] to control somatosensory input during transcranial electrical stimulation, we adopted the same approach. At the beginning of the topical anesthesia experiments, a thick layer of EMLA cream (approximately 3 grams) was applied to a 5-cm (diameter) region surrounding each electrode site. Following the manufacturer's recommendations, the cream was tightly covered with a plastic dressing and allowed to absorb for approximately 1 hour (median: 52 minutes, range: 46–72 minutes). The skin was then cleaned with soap and water, followed by alcohol, and allowed to air-dry before the tACS electrodes were attached. Recording typically began 15–20 minutes later.

We used the animal's behavioral performance to validate the effectiveness of anesthesia. In our previous studies [3, 16], animals were initially distracted and agitated by the onset of tACS but eventually learned to continue working despite the evoked percepts. If EMLA effectively blocks somatosensation, applying it around tACS electrodes should reduce distractions and increase time spent on task. Since naive subjects would be most sensitive to these effects, a well-trained monkey that had never received tACS (Monkey S) was used for this experiment. Two pairs of tACS electrodes were placed on its head, one on anesthetized skin over one hemisphere and the other at identical locations on the untreated contralateral side. After a 20-minute delay that accounted for the setup time needed for neurophysiological experiments, the monkey performed a visual fixation task while bursts of tACS were sporadically applied through each pair of electrodes. As Fig 2 shows, time on task was significantly increased at all stimulation amplitudes when tACS was applied to the anesthetized skin, as compared to control sites (±0.5 mA: $p = 0.028$; Z = −2.19; ±1 mA: $p = 0.009$, Z = −2.60; 2 mA: $p = 0.016$, Z = −2.39, Wilcoxon sign-rank tests).

Since our paradigm depends on tACS-induced sensations being surprising or noxious enough to distract the animal from the fixation task, we cannot completely exclude the possibility that some residual somatosensory input persists, despite the topical anesthesia. However, measurements from human participants, who can provide richer reports of their subjective experiences, find very similar thresholds: ±0.51 mA under control conditions and ±0.93 mA after EMLA topical anesthesia [5]. This suggests that our procedure accurately estimates the monkey's perceptual threshold to be approximately 1 mA.

Experiments were always completed within the 2-hour anesthetic window recommended by the manufacturer (median experiment duration: 68 minutes, range: 24–92 minutes). To verify that the anesthesia remained constant throughout each session, we analyzed each epoch of data collected during anesthesia sessions separately, using a mixed-effects model with fixed effects of stimulation type (tACS or sham), the duration of EMLA pretreatment (in seconds), and the time elapsed between EMLA removal and midpoint of each recording epoch (in seconds). The model also included a random intercept for each epoch (to account for neurons recorded simultaneously) and a random intercept and slope for stimulation type accounting for cell-specific effects. As in the main text, we find that entrainment significantly increases during tACS ($p < 0.001$, t[267] = 3.35), but the model's anesthesia-related coefficients are both indistinguishable from zero (EMLA dose: $p = 0.82$; t[267] = 0.225; time elapsed: $p = 0.49$; t[267] = −0.692). These features had little impact on the model, and removing them increased its parsimony, as measured by AIC and BIC (ΔAIC = 3.5; ΔBIC = 10.7). A similar nonparametric analysis found no significant differences in entrainment between epochs that were recorded 30, 45, or 60+ minutes after EMLA removal ($p = 0.97$; $\chi^2[2] = 0.048$; Kruskal–Wallis test).

## Behavioral task

Since arousal and oculomotor activity can strongly affect neural oscillations, we used a simple fixation task to control the animal's behavioral state and to minimize eye movements. Animals

sat in a standard primate chair (Crist Instruments; Hagerstown, MD, United States), 57 cm from a computer monitor covering the central 30° × 60° of their visual fields. We monitored eye position noninvasively, using an infrared eyetracker (Eyelink-1000; SR Research, Ontario, Canada). Monkeys were trained to fixate a small black target (approximately 0.5°) presented against a neutral gray background. Whenever their gaze remained on this target for 1–2 seconds, they received a liquid reward. Inter-reward intervals were drawn from an exponential distribution (with a flat hazard function) to prevent entrainment to rewards or expected rewards. Both animals had received extensive training before these experiments and tended to maintain their gaze continuously on the fixation target. Custom software written in Matlab (The Mathworks, Natick, MA, USA) controlled the behavioral task and coordinated the eye tracker, tES stimulator, and recording hardware.

### Neural data collection

Single-neuron data were initially obtained from the left hippocampus, an interesting test bed for these experiments because it receives input from a wide range of areas, including sensory ones, yet is not tightly tied to any specific modality. Additional data were collected from area V4, a midlevel visual area that is less coupled to the somatosensory system.

At the start of each recording session, we penetrated the dura with a sharpened 22-ga. stainless steel guide tube. A 32-channel V-Probe with 150-μm spacing (Plexon; Dallas, TX, USA) was then inserted into the guide tube and positioned with a NaN Microdrive (NaN Instruments; Nazareth Illit, Israel). The target depth was determined from the animals' MRIs, and, for monkey N, confirmed via CT, as shown in [3].

Wideband signals were recorded using a Neural Interface Processor (Ripple Neuro; Salt Lake City, UT, USA). Signals were referenced against the guide tube, bandpass filtered between 0.3 and 7,500 Hz, and stored at 30,000 Hz with 16-bit/0.21-μV resolution for offline analysis. The raw wideband signals were first preprocessed with a PCA-based filtering algorithm [37] to attenuate stimulation artifacts. Next, single units were identified by bandpass filtering the signal between 500 and 7,000 Hz with a third order Butterworth filter and thresholding it at ±3 standard deviations. The 2-ms segments around each threshold crossing were then sorted using UltraMegaSort 2000, a *k*-means overclustering algorithm [38]. Its results were manually reviewed and refined to ensure that each unit had a clear refractory period, stable width and amplitude, and good separation in PCA space.

### Data analysis

We quantified the neurons' entrainment to the electrical stimulation using pairwise phase consistency (PPC), a measure of the synchronization between a point process (spikes) and a continuous signal (tACS or LFP) with statistical advantages over a direct calculation of PLVs or spike-field coherence [39]. The PPC scores are an unbiased estimate of the square of the PLVs, a more commonly used measure, so we report values as PLVs to facilitate comparison with other work.

Neurons may fire rhythmically even in the absence of stimulation. We therefore compared entrainment to the tACS waveform (during stimulation) with entrainment to the corresponding frequency component (20 ± 1 Hz) of the LFP, during sham stimulation. Only the middle 4 minutes of each epoch was analyzed, when no current whatsoever was being applied during the sham condition. In both cases, the continuous signal came from an adjacent channel to avoid spectral contamination by the spiking activity [40]. This approach also accounts for physiological distortions of the tACS waveform [41].

Data acquisition or signal processing artifacts could potentially produce the appearance of entrainment. Our previous work [3] describes a number of analyses and controls addressing this issue, using data collected with the same equipment, monkey, and electrodes. We showed that entrainment is unrelated to firing rate or signal amplitude and that neurons' waveforms remain consistent between tACS and sham conditions, as well as across different phases of the tACS. These analyses suggest that the entrainment seen here is unlikely to be due to technical artifacts.

Our analyses combine all data collected in the same condition, though an epoch-by-epoch analysis (see Topical Anesthesia section, above) yields the same conclusion. Where possible, nonparametric tests were used to avoid distributional assumptions. Randomization tests were used to compare PPC values across tACS and sham conditions; population-level analyses were carried using Wilcoxon rank-sum and sign-rank tests, as appropriate, and 95% confidence intervals for the median were calculated using the formula in [42]. All statistical tests are two-tailed, except where noted. Sample sizes were determined based on our previous work and data were analyzed using Matlab (The Mathworks, Natick, MA, USA) and R [43].

## Supporting information

**S1 Fig. Reduced tACS current (±1 mA) still entrains hippocampal neurons.** Data shown in the same style as Fig 3C and 3D. PLVs for individual neurons are indicated by gray points, with lines connecting observations from the same cell. As before, tACS increases neuronal entrainment during TA (green) and control sessions (yellow), compared to the corresponding sham conditions. However, no significant difference (n.s; $p > 0.05$) was detected between tACS and tACS + TA. See S3 Data for individual values of each data point. n.s, not significant; PLV, phase-locking value; TA, topical anesthesia; tACS, transcranial alternating current stimulation.
(TIF)

**S2 Fig. tACS shifts preferred phases to 90˚.** Distribution of preferred firing phases for the neurons shown in Fig 3 during sham (blue) and 20-Hz tACS (yellow) stimulation. During tACS, neurons preferentially fired near the peak of the tACS waveform (90˚), but no such concentration was observed under sham conditions. Note that the preferred phase estimates for sham stimulation are noisy because neurons were minimally entrained to the 20-Hz component of the LFP (Fig 3). See S2 Data for individual values of each data point. LFP, local field potential; tACS, transcranial alternating current stimulation.
(TIF)

**S1 Data. Individual data values for Fig 2.** Each entry indicates the percent of time during which the animal maintained its gaze within 2˚ of the fixation site.
(XLS)

**S2 Data. Individual data values for Fig 3 and S2 Fig.** Rows in red indicate neurons that showed individually significant changes in entrainment, shown as filled shapes in Fig 3A and 3B. TA, topical anesthesia.
(XLSX)

**S3 Data. Individual data values for S1 Fig.** Rows in red indicate neurons that showed individually significant changes in entrainment, shown as filled shapes in Fig 3A and 3B. TA, topical anesthesia.
(XLSX)

## Acknowledgments

We thank Julie Coursol, Cathy Hunt, and Dr. Fernando Chaurand for outstanding technical assistance; Luiza Volpi for assistance during some recordings; and Qiming Cui for discussing and modeling intracranial electric fields.

The views, opinions, and/or findings expressed are those of the authors and should not be interpreted as representing the official views or policies of the Department of Defense or the US Government.

## Author Contributions

**Conceptualization:** Pedro G. Vieira, Matthew R. Krause, Christopher C. Pack.

**Data curation:** Pedro G. Vieira, Matthew R. Krause.

**Formal analysis:** Pedro G. Vieira, Matthew R. Krause.

**Funding acquisition:** Christopher C. Pack.

**Investigation:** Pedro G. Vieira, Matthew R. Krause.

**Methodology:** Pedro G. Vieira, Matthew R. Krause.

**Project administration:** Christopher C. Pack.

**Software:** Pedro G. Vieira, Matthew R. Krause.

**Supervision:** Christopher C. Pack.

**Visualization:** Pedro G. Vieira, Matthew R. Krause.

**Writing – original draft:** Pedro G. Vieira, Matthew R. Krause, Christopher C. Pack.

**Writing – review & editing:** Pedro G. Vieira, Matthew R. Krause, Christopher C. Pack.

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
