## [Editor Report · Decision Letter 0]

31 Mar 2020

Dear Dr Krause,

Thank you for submitting your manuscript entitled "tACS entrains neural activity while somatosensory input is blocked" for consideration as a Short Report by PLOS Biology.

Your manuscript has now been evaluated by the PLOS Biology editorial staff, as well as by an Academic Editor with relevant expertise, and I am writing to let you know that we would like to send your submission out for external peer review.

Please re-submit your manuscript within two working days, i.e. by Apr 02 2020 11:59PM.

Kind regards,

Gabriel Gasque, Ph.D.,

Senior Editor

PLOS Biology

---

## [Decision Letter · Decision Letter 1]

15 May 2020

Dear Dr Krause,

Thank you very much for submitting your manuscript "tACS entrains neural activity while somatosensory input is blocked" for consideration as a Short Report at PLOS Biology. Your manuscript has been evaluated by the PLOS Biology editors, by an Academic Editor with relevant expertise, and by three independent reviewers.

In light of the reviews (below), we are pleased to offer you the opportunity to address the comments from the reviewers in a revised version that we anticipate should not take you very long. We will then assess your revised manuscript and your response to the reviewers' comments and we may consult the reviewers again.

We expect to receive your revised manuscript within 1 month.

**IMPORTANT - SUBMITTING YOUR REVISION**

We are interested in pursuing your manuscript as a *Methods and Resources* article, and not as a Short Report. Please change the article type when resubmitting.

*Resubmission Checklist*

*Published Peer Review*

*PLOS Data Policy*

*Blot and Gel Data Policy*

Sincerely,

Gabriel Gasque, Ph.D.,

Senior Editor

PLOS Biology

REVIEWS:

Reviewer #1: This manuscript addresses the question whether physiological effects of tACS might be due to somatosensory electrical stimulation, and might be seen as a general reply to Asamoah et al. (2019). I have the following comments on the manuscript.

1) In the Transcranial Alternating Current Stimulation subsection of the methods part and referring to Huang et al. (2017), it is stated:

"For the V4 site, a montage consisting of Fp1/P7 was found to produce field of ~1 V/m with 1 mA of stimulating current, which approximates field strengths achievable in humans [21]." This is one of the several references to Huang et al. (2017), and in many of them the feasibility of reaching field strength of 1V/m is claimed.

However, Huang et al. state in their discussion: "Electric field magnitudes in cortex can be as high as 0.2 V/m for 1 mA stimulation current. For typical electrode configurations used in clinical trials maximal field intensity can reach 0.4 V/m when applying 2 mA. More extended areas can reach a value of 0.16 V/m (95th percentile) under 2 mA stimulation. For some electrode montages, current is shunted into deeper areas through highly conductive CSF with maximal values reaching 0.12 V/m. Based on the present data, previous modeling estimates with maximal intensities around 1 V/m and 95th percentile of 0.35 V/m have to be downwards corrected by a factor of two."

From this, I conclude that a field with a peak of about 1V/m requires application of tACS with amplitude of more than 4 mA, which is much bigger than usual tACS intensities. Please clarify this and make sure that the comparison with the achievable intensities in human studies is correct.

2) Related to my previous point, fields produced in usual human tACS studies seem to be 25% of fields generated in the current study (for V4 experiment). How could PLV results of this paper be generalized to human studies? Moreover, could it be that the contribution of peripheral stimulation (compared to the contribution from direct stimulation) be more significant at lower tACS intensities?

3) Due to the electrode positioned at Fp1, I guess retina has received strong stimulation in both hippocampal and V4 experiments. Please discuss if this retinal stimulation, and not direct electrical stimulation of the brain, might have led to the observed results.

4) At the end of the results section, there is an interesting part about the effective phase of tACS. The authors claim that a direct stimulation would result in spikes mainly locked to 90 degrees of the tACS current. What is the logic behind that? As neural membrane integrates the stimulation current, I assume that this phase is dependent on the stimulation current, and increasing the tACS current would decrease this phase towards zero. If this be current, then it would be hard to make conclusions using the preferred tACS phase. Please discuss.

5) From the text, I concluded that the tACS intensity is specified as the amplitude of the sine wave, and not the peak-to-peak. I suggest clearly mentioning this in the Transcranial Alternating Current Stimulation subsection of the methods part to prevent potential confusions.

6) I could not find how the ceiling in Figure 2 is defined. Moreover, as a t-test compares the mean of the population to a hypothetical mean, I could not understand how the t-test for that is applied. Please clarify and mention the details in the manuscript. Please also clarify what each gray dot represents. Does each of them represent the fixation duration for a single trial or some statistical feature of several trials? Please clarify this in the manuscript.

Reviewer #2: Vieira, Krause & Pack investigated the impact of cutaneous somatosensory co-stimulation as alternative explanation for the neuronal entrainment effects observed by tACS in monkeys. They used topical anesthesia (lidocaine) of the skin below the stimulation electrodes to reduce the sensory input originating from peripheral nerves locally innervating the skin, and found no significant reduction in entrainment of neuronal firing by topical anesthesia, while animals appeared less disturbed by the stimulation when topical anesthesia was applied.

Given the ongoing discussion on the effectiveness of tACS in humans and the role of cutaneous peripheral co-stimulation effects (as raised by Asamoah et al. Nat Commun 2019), addressing these issues in non-human primates is particularly important and will be of great value to the wide community using non-invasive brain stimulation techniques in humans. This paper elegantly contributes to a better understanding of the severity of these co-stimulation effects. It is written in a balanced way also addressing the limitations of the presented data (even though I suggest to make some of these limitations even more clear, see comments below).

1. In humans TACS sensations are indeed not necessarily prevented (but only reduced) for higher intensities. While lidocaine removes the sharpness of the sensation and the potential pain, rhythmic cutaneous sensations can still be perceived and reported. How is effective blocking of these rhythmic sensations (which could serve to entrain brain activity) guaranteed in monkeys that cannot even report whether or not the feel something? These painless sensations may not distract the subject but still cause peripheral entrainment.

2. Additional recordings from primary somatosensory cortex (for the hippocampus or V4 targeting montage) would allow to assess the actual effectiveness of the blockade of cutaneous input to S1 during TACS and to compare it with and without topical anesthesia. However, even then, the unintended direct peripheral co-stimulation of cranial nerves (e.g., trigeminal or vagal nerve) or the meninges (both not affected by the superficial topical anesthesia of the skin, but densely innervated) may affect a number of other brain regions (and their networks) without the entrainment necessarily being observable in S1. Especially two-electrode montages such as FP1 -O1/P7 likely target many peripheral neuronal structures that are not located in the skin itself (in humans FP1 is located very closely to the point where the first branch of the trigeminal nerve enters the skull through a small foramen funneling transcranial currents). In summary, an entrainment via peripheral nerve stimulation (not originating from the anesthetized past of the skin and thus equally strong with and without anesthesia) cannot be excluded. This should be made even more clear in the manuscript.

3. The authors used specific montages for stimulating hippocampus or V4, but not the one that was used previously to show the effect on (physiological) tremor entrainment (Asamoah et al.). Maybe the precise montage location (relative to cranial nerves etc.) is indeed crucial for whether or not a strong peripheral co-stimulation mediated entrainment can be observed. So the conclusion of tACS with and without topical anesthesia being equally effective in entraining neurons in the brain has to be limited to the two tested montages.

Reviewer #3: This is an interesting and timely report and the authors provide compelling evidence proving the direct effect of tACS upon cortical neurons. The results clearly indicate that even when somatosensory input is blocked by EMLA cream, 20 Hz tACS was still entraining neural activity, thus providing experimental evidence that tACS reached deep cortical structures such as the hippocampus as well as in more superficial visual area V4. Conclusions are fully supported by the data.The method is sound and the results are clear. I only have a minor comment/question, that is why they only used 20 Hz stimulation to entrain neurons. They refer to their previous work, but my understanding is that 20 Hz might have differential effects compared to 10 Hz, and their effects are often compared in studies with human participants. While the specific effect of frequency of stimulation is not central to the purpose of the study, I was wondering whether different frequencies might affect cortical neurons and/or indirect somatosensory responses differently.

---

## [Decision Letter · Decision Letter 2]

15 Jul 2020

Dear Dr Krause,

Thank you for submitting your revised Methods and Resources paper entitled "tACS entrains neural activity while somatosensory input is blocked" for publication in PLOS Biology. I have now obtained advice from two of the original reviewers and have discussed their comments with the Academic Editor.

We're delighted to let you know that we're now editorially satisfied with your manuscript. However before we can formally accept your paper and consider it "in press", we also need to ensure that your article conforms to our guidelines. A member of our team will be in touch shortly with a set of requests. As we can't proceed until these requirements are met, your swift response will help prevent delays to publication. Please also make sure to address the data and other policy-related requests noted at the end of this email.

*Copyediting*

*Published Peer Review History*

*Early Version*

*Submitting Your Revision*

Sincerely,

Roli Roberts

Roland Roberts PhD

Senior Editor

PLOS Biology

on behalf of

Gabriel Gasque, Ph.D.,

Senior Editor

PLOS Biology

ETHICS STATEMENT:

-- Please include the full name of the IACUC/ethics committee that reviewed and approved the animal care and use protocol/permit/project license. Please also include an approval number.

-- Please include the specific national or international regulations/guidelines to which your animal care and use protocol adhered. Please note that institutional or accreditation organization guidelines (such as AAALAC) do not meet this requirement.

-- Please include information about the form of consent (written/oral) given for research involving human participants. All research involving human participants must have been approved by the authors' Institutional Review Board (IRB) or an equivalent committee, and all clinical investigation must have been conducted according to the principles expressed in the Declaration of Helsinki.

DATA POLICY:

Note that we do not require all raw data. Rather, we ask for all individual quantitative observations that underlie the data summarized in the figures and results of your paper. For an example see here: http://www.plosbiology.org/article/info%3Adoi%2F10.1371%2Fjournal.pbio.1001908#s5

These data can be made available in one of the following forms:

Regardless of the method selected, please ensure that you provide the individual numerical values that underlie the summary data displayed in the following figure: Figures 2, 3A-D, S1, and S2.

Please also ensure that each figure legend in your manuscript include information on where the underlying data can be found and ensure your supplemental data file/s has a legend.

REVIEWERS' COMMENTS:

Reviewer #2:

[identifies himself as Til Ole Bergmann]

The authors adequately addressed all of my previous concerns. I have no further comments.

Reviewer #3:

The authors addressed my minor concern, I am happy with the revision. The full rebuttal is very exhaustive and interesting. Nice work!

---

## [Editor Report · Decision Letter 3]

1 Sep 2020

Dear Dr Pack,

On behalf of my colleagues and the Academic Editor, Simon Hanslmayr, I am pleased to inform you that we will be delighted to publish your Methods and Resources in PLOS Biology.

Early Version

PRESS

Kind regards,

Alice Musson

Publishing Editor,

PLOS Biology

on behalf of

Gabriel Gasque,

Senior Editor

PLOS Biology